# Assessment of Nano-Formulated Conventional Insecticide-Treated Sugar Baits on Mosquito Control and the Effect on Non-Target Aphidophagous *Coccinella septempunctata*

**DOI:** 10.3390/insects15010070

**Published:** 2024-01-18

**Authors:** Muhammad Farhan, Chenchen Zhao, Sohail Akhtar, Ishtiaq Ahmad, Pan Jilong, Shuai Zhang

**Affiliations:** 1College of Plant Protection, Yangzhou University, Yangzhou 225009, China; farhan.entomology@gmail.com (M.F.); mz120231454@stu.yzu.edu.cn (P.J.); 2College of Plant Protection, Henan International Laboratory for Green Pest Control, College of Plant Protection, Henan Agricultural University, Zhengzhou 450002, China; 3Department of Entomology, The Islamia University of Bahawalpur, Bahawalpur 63100, Pakistan; sohail.akhtar@iub.edu.pk; 4Department of Horticultural Sciences, The Islamia University of Bahawalpur, Bahawalpur 63100, Pakistan; ishtiaq@iub.edu.pk

**Keywords:** mosquito, vector-borne diseases, *Coccinella septempunctata*, nano-formulated conventional insecticides, attractive toxic sugar baits

## Abstract

**Simple Summary:**

In this study, we investigated the efficacy of nano-formulated conventional insecticides with attractive toxic sugar baits (N-ATSB) in comparison to their traditional counterparts, commonly used for both agricultural pest and mosquito control, and also observed the non-target effect on *Coccinella septempunctata*. Ten conventional insecticides were employed, administered both in traditional form and as nano-particles with ATSB. The study focused on mosquito strain *Anopheles gambiae* collected from various crop fields, testing mortality rates through adult bioassays at 36 and 72 h. Notably, the slow response of certain insecticides resulted in higher mortality after 72 h. Significant differences in mosquito mortality were observed among all tested insecticides, particularly with high (1%) and low (0.5%) doses of both traditional and nano-formulated versions. Applying ATSB solution with each insecticide showcased a marked impact on mosquito mortality. The results revealed variations in efficacy among different insecticides, with carbosulfan (nano-formulation) demonstrating the highest toxicity, recording 98% mortality. The nano-formulated conventional insecticides showed no adverse effects on the ladybird beetle compared to their commercial counterparts. This study provides valuable insights to improve mosquito control strategies while reducing the impact on non-target organisms. It highlights the potential of N-ATSB in enhancing the effectiveness of insecticides against disease vectors.

**Abstract:**

Mosquitoes, as disease vectors causing global morbidity and mortality through diseases like malaria, dengue, and Zika, necessitate mosquito population control methods. This study investigated the efficacy of nano-formulated insecticide-based sugar baits in controlling *Anopheles gambiae* populations and assessed their potential non-target impact on *Coccinella septempunctata*. This laboratory-based study employed thiolated polymer-coated attractive toxic sugar bait (ATSB) nano-formulations, delivering pesticides via nano-carriers. Adult and larvae populations of insects were collected from rice and cotton fields subjected to bioassays with 0.5% and 1% concentrations of each nano-formulated and conventional insecticide within ATSB solution, alongside a control 100% attractive sugar bait (ASB). Mosquitoes interacted overnight with insecticide-treated baits, and mortality was assessed. Further observations up to 72 h were conducted for potential delayed toxic effects. Results highlighted nano-ATSB carbosulfan’s effectiveness, particularly among organophosphates and pyrethroids. Among pyrethroids, nano-ATSB cypermethrin exhibited high efficacy, while Deltamethrin displayed lower mortality. Among organophosphates, nano-ATSB chlorpyrifos induced substantial mortality. The nano-formulations of insecticide were harmless against *C. septempunctata* compared to their conventional form. Nano-formulations demonstrated enhanced mortality rates and prolonged efficacy against mosquitoes, having a benign impact on non-target beetles. We expect these results to aid in developing effective plant protection products suitable for IPM practices.

## 1. Introduction

Mosquitoes are a significant concern for human health worldwide, as they serve as vectors for transmitting dangerous diseases [1,2]. The mosquito family (*Culicidae*) of arthropods is responsible for transmitting many of the world’s deadliest human diseases. Malaria, spread by *Anopheles* mosquitoes [3], causes illness in over 200 million people annually and kills between 6 and 12 million people, primarily children in low-income countries [4]. The mosquitoes of the genus *Aedes*, which are easily identified by their striking black and white markings, were responsible for spreading the yellow fever virus that caused devastating epidemics from Washington, DC, in the New World to Barcelona, Spain, in the Old World in the 18th and 19th centuries [5]. Dengue fever is the most dangerous disease transmitted by *Aedes*, expanding quickly throughout the tropics and semi-tropics [6,7]. Blood-feeding activity among this species peaks at the exact times of day as humans do, namely early morning and late afternoon [8]. *Ae. aegypti*, prefer to feed and relax in enclosed spaces, often adjacent to their breeding sites [8,9].

Nematodes (*Wuchereria*, *Brugia*), which are responsible for filarial elephantiasis, are transmitted to humans through the bites of certain mosquito species [10]. A specific group of arboviruses is poorly characterized because they primarily circulate in animal hosts and are rarely transmitted to humans without being vectored between humans. Livestock and wild animals are susceptible to mosquito-borne illnesses. For instance, horses in the United States can contract equine encephalitis caused by various viruses [11]. Native Hawaiian bird populations have been in steep decline owing to the introduction of insects and avian malaria [12]. Finally, it has recently come to light that mosquito-borne infections may have a more vital link to cancer than previously thought [13] and that the immune system has problems recognizing and fighting specific pathogens [14].

Synthetic pesticides have become increasingly crucial in vector control operations, particularly during disease outbreaks [15]. Pesticide resistance and its associated environmental damage have emerged in recent years due to increased pesticide use [16]. Most currently available pesticides are neurotoxic, disrupting insect nervous systems, which may also harm mammals and non-target insects, such as pollinators [17]. Diseases related to oxidative stress are linked to improper use of pesticides in agriculture, which alters the balance between antioxidant and oxidant enzymes in the human body [18]. To this end, it is crucial to identify novel approaches considering the natural environment [16,17,18,19]. Nano-technology has shown promise in creating insecticides with less environmental impact to bridge this gap [20,21,22,23]. The medical and veterinary sciences [24] and entomology [25] are only two examples of the many areas of study that have benefited from nano-technology in recent years. Nano-pesticides have the potential to encompass a wide variety of pesticide formulations that have proven helpful as carriers of potent pesticides over the past decade [26]. Because of their high surface-area-to-volume ratio, these formulations are more water soluble than the excited insecticide [27].

Research is being conducted into novel mosquito control technologies that meet the WHO guidelines, such as mosquito baiting, to combat the existing mosquito control issues. It was found that mosquito baiting tactics benefit from considering mosquito biology, mosquito ecology, and mosquito behavior [28]. The attractive toxic sugar bait (ATSB) method is helpful for mosquito control since both male and female mosquitoes need sugar meals to survive [29]. *Culicidae*, a family of insect vectors, have been investigated for their sugar-feeding habits [30]. Mosquitoes can identify sources of sugar by seeing or smelling them or touching their tarsi. Eating is induced by contact with sugars. Thus, the circadian rhythm of mosquitoes may control flower selection. To survive, adult mosquitoes require a sugar diet soon after they emerge from their larval stage. Without carbohydrate stores, the odds of mosquito mating, blood feeding, growth, and egg production are low [31]. Since mosquitoes primarily feed on liquids, a novel approach to mosquito control has been proposed to incorporate stomach toxins into a feeding stimulant or arrestant and distribute it in areas where mosquitoes rest, such as larval habitats and foliage near host habitats [32]. After finding an ATSB, mosquitoes are killed when they drink the poisonous fluids [33].

It has been established that outdoor mosquito populations can be reduced using ATSB techniques. It seems plausible to test whether or not they are also successful in reducing mosquito populations inside [34]. Using sugar baits has reduced mosquito populations and had negligible effects on other arthropods. Toxic sugar baits (TSB) and ATSBs are an effective form of mosquito control [28]. The current research tested an alternative, eco-friendly approach to mosquito control using nano-technology and attractive toxic sugar baits.

The application of synthetic insecticides for the chemical control of mosquitoes in recent decades has led to the emergence of insecticide resistance and has also had detrimental effects on natural predators, such as ladybird beetles [35]. The aphidophagous seven-spotted ladybird beetle *Coccinella septempunctata* (Coleoptera: Coccinellidae) plays a vital role in biological control and IPM [36]. These essential ecosystem components have been utilized globally to manage pests like mites, mealy bugs, aphids, and thrips. Thus, researchers have given this bio-agent a lot of attention. Since this insect has become crucial to modern agriculture, great efforts are being made to shield it from exposure to dangerous chemicals [37]. Natural enemies are thought to have a significant role in insect population control. The most severe impediment to fulfilling the potential of natural enemies in field crops is disruption caused by the extensive use of insecticides with broad toxicity to both pests and natural enemies [38]. Considering nano-technology as a new tool to enhance the profile and activity of pesticide formulations [39], we assessed the comparative effectiveness and potential adverse effects of some commercial and nano-formulated conventional insecticides treated with ATSB. For this purpose, we evaluated the insecticidal activity of nano-formulations in a laboratory bioassay against *An. gambiae*. Additionally, since it is essential to safeguard natural predators of aphids in agricultural settings, we checked for any possible harm to non-target predators like *C. septempunctata*.

## 2. Materials and Methods

### 2.1. Preparation of Attractive Sugar Baits (ASB)

The ASB solution used ripe fruit juice, which is known to attract mosquitoes. It was made by mixing guava juice and sucrose in a 1:1 ratio. The juice was also ripened at ambient temperature for 48 h in a closed container before scrambling the solution [40]. This combination created a sweet and appealing bait that attracted mosquitoes, helping control their population naturally and safely.

### 2.2. Preparation of Attractive Toxic Sugar Baits (ATSB)

To prepare attractive toxic sugar baits (ATSB), 1% boric acid was used. For this purpose, 1% boric acid was mixed in a 99% solution of attractive sugar baits.

### 2.3. Preparation and Characterization of Conventional Insecticides Nano-Formulations

Preparation and characterization of conventional insecticide nano-formulations were conducted by the Department of Pharmacy Quaid-i-Azam University Islamabad according to their developed protocol. Briefly, 0.2% (*w*/*v*) of thiolated chitosan suspensions were prepared in 1% acetic acid. Sodium tripolyphosphate (TPP, 1.0%), of which conventional insecticides are comprised, was added dropwise to 6 mL of chitosan with stirring, followed by sonication for 10 s. The resulting thiolated chitosan particle suspension was centrifuged at 10,000× *g* for 10 min. The pelleted particles were resuspended in deionized water with 10-s sonication and lyophilized. The thiolated chitosan nano-particles’ particle size, zeta potential, and PDI (polydispersity index) were determined using zeta nano-sizer ZSP (Malvern, UK). The recorded data revealed a particle size of 98.024 nm, a PDI of 0.158, and a ZSP of −24.3 mV. One optimized formulation of each conventional insecticide was chosen based on particle size, zeta potential, polydispersity index (PDI), and entrapment efficacy for further experiments.

### 2.4. Preparation of Nano-Formulated Attractive Toxic Sugar Baits (N-ATSB)

Similarly, nano-formulated attractive toxic sugar baits (N-ATSB) were prepared by adding the nano-formulated conventional insecticides to the attractive toxic sugar baits. Two concentrations (0.5% and 1%) of each nano-formulated conventional insecticide were used to form a hundred percent N-ATSB solution for laboratory bioassays.

### 2.5. Preparation of ATSB of Insecticides in Conventional/Traditional Form

Conventional insecticides were also used in their synthetic simple form with attractive toxic sugar baits (ATSB), to evaluate the comparative effectiveness of simple/traditional and nano-formulated insecticides with ATSB solution. Like nano-formulations, the same two concentrations (0.5% and 1%) were used. This formulation was named ATSB of insecticides in conventional form.

### 2.6. Study Design

The insecticides examined were categorized into three groups: pyrethroids (lambda-cyhalothrin, cypermethrin, deltamethrin, bifenthrin), organophosphates (triazofos, profenofos, chlorpyrifos), and carbamates (carbosulfan, propoxur, methomyl). Each group experienced consistent concentration treatments. The control treatment (T_1_) included ASB (100%). Treatments T_2,_ T_3_, T_4_, and T_5_ incorporated insecticides with ATSB, maintaining consistent concentrations. T_2_ featured a 0.5% insecticide concentration + 99.50% ATSB, T_3_ utilized a 1% insecticide concentration + 99% ATSB, T_4_ employed a 0.5% nano-formulated insecticide concentration + 99.50% ATSB, and T_5_ applied a 1% nano-formulated insecticide concentration + 99% ATSB. Notably, the study compared conventional and nano-formulated insecticide forms using ATSB as the solution.

### 2.7. Field Collection and Stock Culture of Mosquito

Two mosquito sample collection and preparation techniques were used at each study site: (1) collection of adult mosquitoes from different crop fields; and (2) collection of mosquito larvae from the trials installed in different rice and cotton fields. For the first method, adult mosquitoes were collected using an aspirator in different fields, especially from rice fields. Rice fields are principal breeding grounds for mosquitoes. About 40 different mosquito species can be found breeding in rice paddies; however, *An. Gambiae* adults were identified through taxonomic key [41]. These adults were transferred to the cage in the Entomology Research Laboratory. These were immediately provided with 10% sucrose solution as food and blood sources. The eggs and larvae were collected from the installed trials and stagnant water sump near the Department of Agriculture for the second method. They were collected through a spoon and transferred to a small plastic box. A large number of larvae and eggs were collected daily. These were also moved to the Entomology Research Laboratory in the cage/box with a proper ventilation system. The Laboratory room has a 12-h day/night schedule and is kept at a constant 28 °C and 80% humidity.

### 2.8. C. septempunctata Collection and Rearing

The adults of seven-spotted ladybird beetles (*C. septempunctata* L.) were collected from cotton fields during spring 2022. The insect population was maintained on cotton aphids *Aphis gossypii* (Hemiptera: Aphididae) in the lab. The beetles were exposed to aphids in cylindrical glass cages on infested cotton plants at 25 ± 1 °C, 65 ± 5% RH, and a photoperiod of 16:8 (L:D) hours.

### 2.9. Mosquito Adult Bioassay

Bioassays were conducted on (F1 generation) 3–5-day-old *An. Gambiae* strains (50 adults) using the precise ATSB with insecticides in conventional form and N-ATSB solution that prevents excessive dripping when absorbed onto the cotton wool pad. The bait was deposited in a petri dish on the floor of the mosquito breeding cage (40 cm^3^) and was replaced daily. Before laboratory experiments, mosquitoes were starved for six hours by removing sugar sources from the testing enclosures. Two different concentrations, 0.5 and 1% of each conventional insecticide and nano-formulated conventional insecticides separately with ATSB (attractive toxic sugar baits) to make a hundred percent solution, were tested in succession followed by 3–4 replications. ASB (without insecticide) was used as a control. Mosquitoes were left overnight to interact with the ATSB and N-ATSB of insecticides and mortality was measured after 36 and 72 h due to the potential slow toxicity of the insecticides. Before every bioassay replicate, the metal enclosures were replaced to prevent sugar contamination.

### 2.10. C. septempunctata Bioassay

The same concentrations of the insecticides were sprayed by Potter Precision Laboratory Spray Tower on the cotton leaves and left to dry. The bioassay was conducted using Scintillation glass vials (30 mL). To check the non-target effect of the commercial and nano-formulated conventional insecticides, ten adults were transferred to each glass vial with three replications of each insecticide. The cotton aphids were provided as food for predatory beetles. The treated glass vials were closed with cotton plugs, and the mortality was recorded after 36 and 72 h of exposure.

### 2.11. Statistical Analysis

In the present study, statistical analysis was employed to rigorously assess the efficacy of nano-formulated conventional insecticide-treated sugar baits for mosquito control and non-target effect on *C. septempunctata*. Statistical analyses were subjected to arcsine square root transformation in SPSS to attain normality; the general linear model (GLM), followed by Tukey’s pair-wise comparison, was used to evaluate the significance among the percentage mortality rates.

## 3. Results

### 3.1. Comparative Efficacy of Conventional and Nano-Formulated Pyrethroid-Treated ATSB

Figure 1 represents the comparative efficacy of some commonly used pyrethroid insecticides: lambda-cyhalothrin, cypermethrin, deltamethrin, and bifenthrin. The results showed that at T_5,_ after 72 h of exposure (combination of 1% nano-formulated lambda-cyhalothrin and 99% of ATSB), highest mortality (84%) was recorded, while at T_3_ (combination of 1% conventional lambda-cyhalothrin and 99% of attractive toxic sugar baits), the lowest mortality (72.70%) was noticed, which is lower than T_4_ (79%) (*p* < 0.001), where less concentrated 0.5% nano-formulated lambda-cyhalothrin with 99.50% ATSB was used. Moreover, T_1_ represents the controlled treatment in which a 100% prepared solution of ASB was used without insecticide (Figure 1a). The results show that at T_5_ N-ATSB, cypermethrin exhibited more significant mortality (87.30%) using the same concentrations as the previous one. The conventional formulation of cypermethrin at a concentration of 1%, combined with 99% ATSB, exhibited a peak mortality rate of 75.7%. Evidently, at T_4_, the nano-formulated cypermethrin demonstrated higher insecticidal efficacy, utilizing a lower concentration than the conventional cypermethrin at T_3_, which required a higher concentration for similar results (Figure 1b).

### 3.2. Comparative Efficacy of Conventional and Nano-Formulated Organophosphate-Treated ATSB

The findings indicate that at T_5_ (a combination of 1% nano-formulated triazofos and 99% of ATSB), the highest mortality, 85.30%, was recorded. In comparison, at T_2_ (a combination of 0.5% conventional triazofos and 99.50% attractive toxic sugar baits), the lowest mosquito mortality, 68.30%, was observed. However, the highest insecticidal activity of triazofos in the conventional form, 75.30%, was noticed at T_3_ after 72 h, while the nano-formulated triazofos showed more mortality with low concentration (Figure 2a). The N-ATSB application of nano-formulated profenofos reveals the highest recorded mortality rate of 76.33% with a combination of 1% nano-formulated profenofos and 99% ATSB (Figure 2b). In the case of chlorpyrifos, the graphical representation depicting the comparative efficacy between the N-ATSB application of chlorpyrifos and the ATSB application of conventional chlorpyrifos indicates that the highest recorded mortality of 91.67% was achieved at the specified concentration (a combination of 1% nano-formulated chlorpyrifos and 99% ATSB). The results demonstrate that the N-ATSB application of chlorpyrifos is more effective against *An. gambiae* than the other mentioned organophosphates (Figure 2c). 

### 3.3. Comparative Efficacy of Conventional and Nano-Formulated Carbamate-Treated ATSB

The comparative graphical analysis of the N-ATSB application of carbosulfan and the conventional form of carbosulfan using ATSB shows that at T_5_ (comprising a combination of 1% nano-formulated carbosulfan and 99% ATSB), the highest mortality of 98.30% was achieved. Conversely, at T_3_, the conventional carbosulfan application (1%) with 99% ATSB solution resulted in a maximum mortality of 85.30% (Figure 3a). Furthermore, the nano-formulated carbosulfan (0.5%) exhibited more insecticidal activity than the conventional form. Again, nano-formulated propoxur also showed remarkable insecticidal activity, resulting in a mortality rate of 93.30% using the same concentration against *An. gambiae* (Figure 3b). Methomyl, the third most commonly employed carbamate for different agricultural pests, demonstrated a mortality rate of 89.70%. The findings indicated that methomyl’s N-ATSB application exhibited lower toxicity than the other utilized carbamates. After a comprehensive analysis of all the results, it can be confidently concluded that the N-ATSB application of carbosulfan (a carbamate) demonstrated the highest mortality (98.30%) against *An. gambiae* (Figure 3c).

### 3.4. Comparative Effect of Commercial and Nano-Formulated Conventional Insecticide-Treated sATSB on C. septempunctata

Our results demonstrated a time effect in all pesticide treatments, and the mortality rate at 72 h was significantly higher than that at 36 h. The commercial form of cypermethrin in pyrethroids showed maximum mortality (90%) after 72 h of predatory beetle *C. septempunctata* exposure. In contrast, the nano-formulated cypermethrin showed 15.67% mortality at the same concentration (*p* < 0.05). Lambda-cyhalothrin exhibited 76.00% and 81.67% mortality after 36 and 72 h, respectively, in conventional form, while 12.67% (*p* < 0.05) and 14.33% (*p* < 0.05) mortality were recorded after 36 and 72 h of exposure, respectively, in the nano-formulated form of the insecticides treated with attractive toxic sugar baits. Deltamethrin and bifenthrin showed considerably similar insecticidal activity against the beetle. However, it was noticed that bifenthrin and deltamethrin were less effective against mosquito control, while these insecticides drastically affected the predatory beetle (Table 1).

In the case of organophosphates, the conventional form of chlorpyrifos caused 80.33% and 94.33% mortality against *C. septempunctata* after 36 and 72 h; however, using the same concentration in nano-form, 13% and 20% mortality were recorded after 36 and 72 h (*p* < 0.05), respectively. After chlorpyrifos, profenofos exhibited 92% mortality after 72 h, while triazofos caused 80.66% mortality in the commercial form and 16.67% and 13.67% mortality in the nano-form of these insecticides after 72 h of exposure (*p* < 0.05), respectively (Table 2).

The results showed that carbosulfan displayed the maximum insecticidal activity (96%) against the *aphidophagous* beetle, whereas 22.67% mortality was noticed in nano-form. Propoxur (88.33%, 15%) and methomyl (87.33%, 19%) showed significantly similar insecticidal activity in commercial form but different activity in nano-formulation of insecticides treated with ATSB after 72 h (Table 3). T_1_ was kept as a control, and only ASB was used without insecticide. No mortality was recorded because *C. septempunctata* is attracted to sugar and feeds on plant pollen and honeydew nectar. It was concluded from the results that nano-formulated insecticides treated with ATSB were most effective against mosquitoes and had less harmful effects on non-target organisms like predatory beetles.

## 4. Discussion

This study investigates the potential synergistic effects of combining nano-formulated conventional insecticides with sugar baits to enhance mosquito control and to minimize the environmental toxicity and impact on non-target organisms. Pyrethroids are the most widely used traditional category of insecticides, seeing widespread application in both consumer and public health settings. This study used different pyrethroids (N-ATSB of lambda-cyhalothrin, cypermethrin, deltamethrin, and bifenthrin) as adulticides against *An. gambiae*. The author of [42] evaluated the efficacy of different pyrethroid insecticides against some mosquito species. The results indicated that cypermethrin showed a more significant percent mortality, followed by lambda-cyhalothrin and deltamethrin. Similarly, the findings of that study in the conventional form of insecticides with ATSB solution and with nano-formulation (N-ATSB) revealed the same insecticidal efficacy against the mosquito. Among all the pyrethroid insecticides used, ATSB and N-ATSB of cypermethrin showed the highest mortality, while N-ATSB of deltamethrin showed the lowest mortality. However, ATSB and N-ATSB of lambda-cyhalothrin and bifenthrin indicated approximately equal mortality. Moreover, the findings of that study demonstrate that these nano-formulations of the insecticides treated with ATSB showed more significant mortality even at low concentrations. Similarly, it discovered that lambda-cyhalothrin nano-particles could be an effective larvicide against *Culex pipeins* larvae at reduced concentrations, thereby eliminating the various drawbacks of the conventional version [43]. The effectiveness of cypermethrin was compared with environmentally friendly biopesticides for managing *Anopheles stephensi* mosquitoes, revealing significant mortality rates [44].

Although pyrethroids are the only type of insecticide allowed in insecticide-treated nets, serious concerns exist about using them for indoor residual spraying due to their low cost and long shelf life. In addition, pyrethroid resistance is expected since this kind of insecticide is widely used in insecticide-treated nets and because its action methods are similar to those of organochlorines. Only carbamates and organophosphates would remain effective options for mosquito control, which is problematic due to their higher cost and the possibility that they share a resistance mechanism (insensitive acetylcholinesterase) [45]. 

Carbamates and organophosphates are the most common alternatives to pyrethroids for mosquito control, and they are effective against both pyrethroid-resistant and -susceptible *Culex* and *Anopheles* populations [46]. When tested against wild mosquitoes in experimental huts, organophosphates and carbamates were found to be non-excitorepellent and very active at killing *Anopheles*, *Culex*, and *Mansonia* [47]. However, using carbamate organophosphates against mosquitoes raises concerns about human safety because of their role as potent cholinesterase inhibitors [48]. One of the primary goals of this study is to reduce the risk to human health. These nano-formulated insecticides worked well to control malarial vectors because they were effective at low doses, safe for people and non-target organisms, had excitorepellent properties, killed insects quickly, and kept killing them for a long time. In the previous literature, the same results were observed against mosquitoes, but these insecticides were used in their conventional forms, posing environmental and human health risks. Among all the pyrethroids used, N-ATSB of chlorpyrifos (organophosphates) showed the highest percentage mortality against *An. gambiae*. It was discovered that chlorpyrifos were effective against pyrethroid-resistant anophelines, cyanines, and other organophosphates used in the net experiments [49]. Similar results were observed in the current study, as organophosphates were more effective than pyrethroids. 

Tested bendiocarb and pyrethroid-treated curtains against *Culex quinquefas ciatus*, the vector of bancroftian filariasis, and found that the former performed better than the latter [50]. The manufacturer deemed the carbamate too harmful for bed nets [51].

Carbosulfan, a carbamate, was more efficient than pyrethroids against *An. gambiae* and to kill both pyrethroid-susceptible and pyrethroid-resistant strains of *An. stephensi* at lower doses. To prevent *An. gambiae* from entering homes, spraying curtains with carbosulfan is beneficial [52]. Using carbosulfan-treated nets in the field showed promising results against pyrethroid-resistant *An. gambiae* and *Cx. quinquefasciatus* [46,47,48,49,50,51,52,53]. Similarly, the current study used the carbamates in conventional and nano-formulation and treated them with attractive sugar baits. The research also revealed that ATSB and N-ATSB of carbosulfan possess the highest insecticidal activity against *An. gambiae* as compared to other insecticides used. The highest increased mortality, 98%, was recorded even at lower concentrations. Additionally, propoxur has been in use since 2012. We evaluated the residual efficacy of propoxur for malarial control and analyzed that propoxur showed the best insecticidal activity against malaria in Ethiopia [44]. The residual effectiveness of two insecticides was assessed on various substrates over six months. The study revealed that bendiocarb displayed a brief residual efficacy, whereas propoxur exhibited more prolonged residual efficacy, leading to over 80% mosquito mortality [54]. The current study’s results also match the results of [55]. Propoxur in conventional form with ATSB almost showed the same insecticidal activity (79.30%), while nano-formulation treated with ATSB showed more effectiveness.

Our study also revealed that the nano-formulations of conventional insecticides showed low toxicity on the non-target predatory beetle *C. septempunctata*. The efficacy of commercial pyrethroids as insecticides exceeds that of nano-formulations. However, it is crucial to note that these commercial pyrethroids are classified as harmful to aphids’ natural enemies, with a significant impact observed on ladybird beetles [55,56]. Our results showed that in pyrethroids, cypermethrin in conventional form caused the highest residual activity against mosquitoes and the non-target predatory beetle *C. septempunctata*. Similarly, a study published similar results of cypermethrin in commercial form against the ladybird beetle [57]. In the case of organophosphates, chlorpyrifos exhibited the maximum insecticidal activity and caused 94.33% mortality in commercial form, while 20% mortality was recorded in nano-formulation treated with ATSB. A hundred percent mortality of predatory beetles was observed using chlorpyrifos after 72 h of exposure under laboratory conditions [58]. Our results did not match this study because of the attractive toxic sugar baits and concentration difference. We evaluated the efficacy of profenofos against the aphidophagous beetle and recorded similar results [59]. In the case of carbamates, carbosulfan was the most effective insecticide against mosquitoes and caused the highest mortality of ladybird beetles in commercial form. Similar outcomes were observed in a field study on brinjal, where the application of carbosulfan to control the jassid attack exhibited comparable effects on ladybird beetles [60]. The increased insecticidal effectiveness of nano-formulated conventional insecticides and their non-toxic impact on non-target aphid predators establish them as compatible plant protection options for organic farming and integrated pest management (IPM) across diverse crops. These results further support nano-technology applications in optimizing insecticide formulations to develop effective and environmentally friendly plant protection products.

## 5. Conclusions

The study developed and tested ATSB with conventional insecticides in traditional and nano-particle forms, including carbamates, pyrethroids, and organophosphates. The results indicate that N-ATSB (nano-attractive toxic sugar bait), formulated with carbosulfan, a carbamate, demonstrated the highest mortality rates among all tested insecticides. It achieved a 98% mortality against *An. gambiae* and recorded a significant 22.67% mortality on non-target predatory *C. septempunctata*. N-ATSB of cypermethrin and permethrin were the most effective among the pyrethroids. At the same time, N-ATSB of chlorpyrifos demonstrated the highest mortality among the organophosphates, reaching up to 91.70%. However, it was concluded that nano-formulated conventional insecticides had minimal adverse effects on non-target aphidophagous *C. septempunctata*.

## Figures and Tables

**Figure 1 insects-15-00070-f001:**
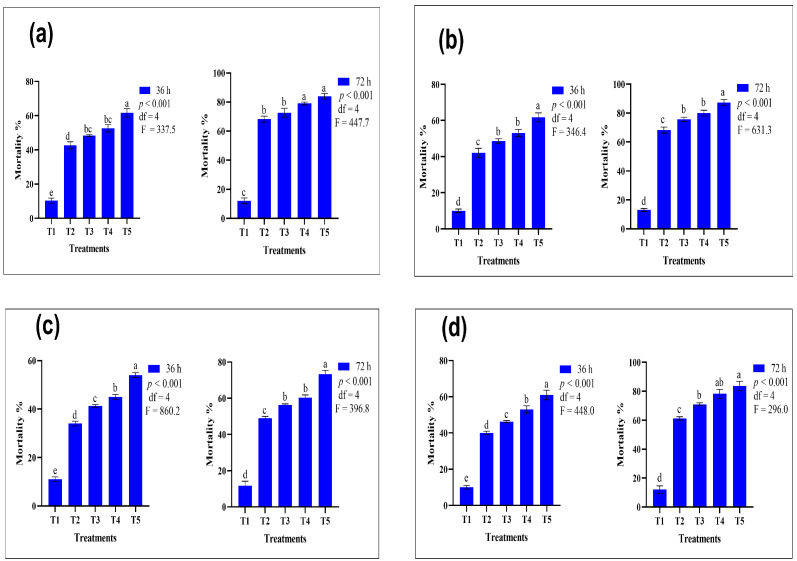
Mean mortality comparison of N-ATSB and ATSB pyrethroid applications against *An. gambiae*: (**a**) lambda-cyhalothrin, (**b**) cypermethrin, (**c**) deltamethrin, (**d**) bifenthrin. The graphs’ bars represent the mean values of three replicates, and lowercase letters denote significant differences between treatments at *p* < 0.05. T_1_ included ASB (100%), T_2_ featured a 0.5% insecticide concentration + 99.50% ATSB, T_3_ utilized a 1% insecticide concentration + 99% ATSB, T_4_ employed a 0.5% nano-formulated insecticide concentration + 99.50% ATSB, and T_5_ applied a 1% nano-formulated insecticide concentration + 99% ATSB.

**Figure 2 insects-15-00070-f002:**
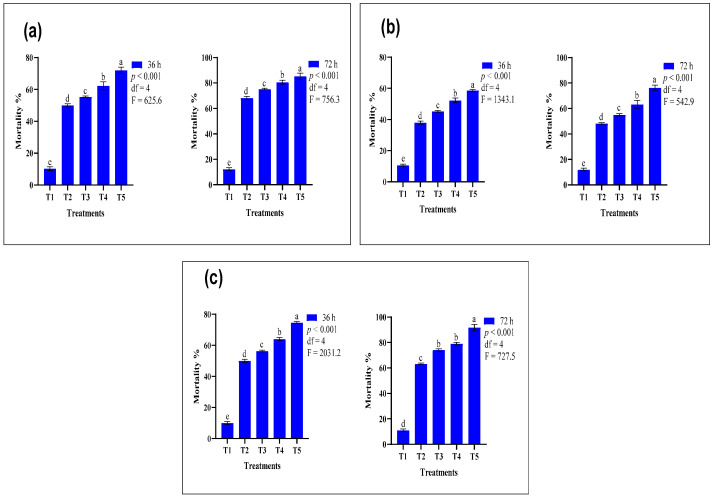
Mean mortality comparison of N-ATSB and ATSB organophosphate applications against *An. gambiae*: (**a**) triazofos, (**b**) profenofos, (**c**) chlorpyrifos. The graphs’ bars represent the mean values of three replicates, and lowercase letters denote significant differences between treatments at *p* < 0.05. T_1_ included ASB (100%), T_2_ featured a 0.5% insecticide concentration + 99.50% ATSB, T_3_ utilized a 1% insecticide concentration + 99% ATSB, T_4_ employed a 0.5% nano-formulated insecticide concentration + 99.50% ATSB, and T_5_ applied a 1% nano-formulated insecticide concentration + 99% ATSB.

**Figure 3 insects-15-00070-f003:**
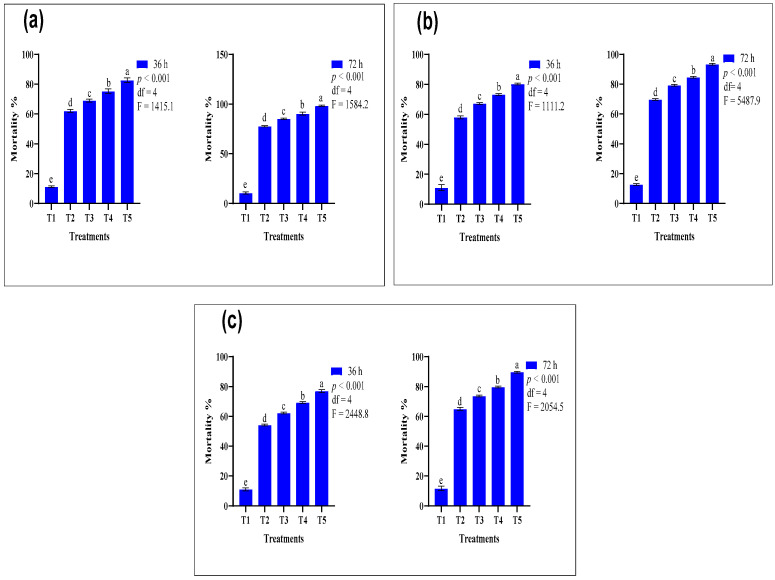
Mean mortality comparison of N-ATSB and ATSB carbamate applications against *An. gambiae*: (**a**) carbosulfan, (**b**) propoxur, (**c**) methomyl. The graphs’ bars represent the mean values of three replicates, and lowercase letters denote significant differences between treatments at *p* < 0.05. T_1_ included ASB (100%), T_2_ featured a 0.5% insecticide concentration + 99.50% ATSB, T_3_ utilized a 1% insecticide concentration + 99% ATSB, T_4_ employed a 0.5% nano-formulated insecticide concentration + 99.50% ATSB, and T_5_ applied a 1% nano-formulated insecticide concentration + 99% ATSB.

**Table 1 insects-15-00070-t001:** Effect of commercial and nano-formulated pyrethroids on non-target adult *Coccinella septempunctata*.

Insecticides	Treatments	Mortality after 36 h (%)	Mortality after 72 h (%)	GLM Test Results
Lambda-cyhalothrin	T_1_	0	0	Treatment: F = 2678.9, *p* < 0.001 Time: F = 66.7, *p* < 0.001 Treatment × time: F = 9.4, *p* < 0.001
	T_2_	67.67 ± 2.05 ^b^	81.00 ± 2.94 ^a^	
	T_3_	76.00 ± 1.41 ^a^	81.67 ± 2.36 ^a^	
	T_4_	9.33 ± 0.47 ^d^	13.67 ± 0.47 ^b^	
	T_5_	12.67 ± 0.47 ^c^	14.33 ± 0.47 ^b^	
Cypermethrin	T_1_	0	0	Treatment: F = 1294.1, *p* < 0.001 Time: F = 21.3, *p* < 0.001 Treatment × time: F = 6.3, *p* < 0.001
	T_2_	75.00 ± 4.08 ^a^	84.00 ± 1.41 ^a^	
	T_3_	79.33 ± 0.94 ^a^	90.00 ± 4.08 ^a^	
	T_4_	13.67 ± 0.47 ^b^	14.67 ± 0.47 ^b^	
	T_5_	15.00 ± 0.82 ^b^	15.67 ± 0.47 ^b^	
Deltamethrin	T_1_	0	0	Treatment: F = 1973.3, *p* < 0.001 Time: F = 52.1, *p* < 0.001 Treatment × time: F = 8.6, *p* < 0.001
	T_2_	56.67 ± 2.36 ^b^	70.67 ± 0.94 ^a^	
	T_3_	65.00 ± 4.08 ^a^	73.33 ± 2.36 ^a^	
	T_4_	7.33 ± 0.94 ^c^	9.67 ± 0.47 ^b^	
	T_5_	8.67 ± 0.47 ^c^	10.67 ± 0.47 ^b^	
Bifenthrin	T_1_	0	0	Treatment: F = 1682.2, *p* < 0.001 Time: F = 37.8, *p* < 0.001 Treatment × time: F = 6.2, *p* < 0.001
	T_2_	63.33 ± 2.36 ^a^	76.33 ± 3.09 ^a^	
	T_3_	70.00 ± 4.08 ^a^	78.00 ± 1.41 ^a^	
	T_4_	7.67 ± 0.94 ^b^	10.00 ± 0.82 ^b^	
	T_5_	9.00 ± 0.82 ^b^	11.00 ± 0.82 ^b^	

T_1_ included ASB (100%), T_2_ featured a 0.5% insecticide concentration + 99.50% ATSB, T_3_ utilized a 1% insecticide concentration + 99% ATSB, T_4_ employed a 0.5% nano-formulated insecticide concentration + 99.50% ATSB, and T_5_ applied a 1% nano-formulated insecticide concentration + 99% ATSB. For each insecticide, means ± standard error followed by different lowercase letters (a,b,c,d) in a column indicate significant differences between treatments (*p* < 0.05).

**Table 2 insects-15-00070-t002:** Effect of commercial and nano-formulated organophosphates on non-target adult *Coccinella septempunctata*.

Insecticides	Treatments	Mortality after 36 h (%)	Mortality after 72 h (%)	GLM Test Results
Triazofos	T_1_	0	0	Treatment: F = 2448.7, *p* < 0.001 Time: F = 57.4, *p* < 0.001 Treatment × time: F = 6.3, *p* < 0.001
	T_2_	66.00 ± 1.41 ^a^	76.33 ± 1.70 ^a^	
	T_3_	71.67 ± 2.36 ^a^	80.67 ± 2.49 ^a^	
	T_4_	8.33 ± 1.25 ^b^	11.33 ± 0.94 ^b^	
	T_5_	10.67 ± 1.25 ^b^	13.67 ± 0.47 ^b^	
Profenofos	T_1_	0	0	Treatment: F = 3183.2, *p* < 0.001 Time: F = 153.4, *p* < 0.001 Treatment × time: F = 21.0, *p* < 0.001
	T_2_	73.67 ± 1.70 ^a^	85.67 ± 1.70 ^b^	
	T_3_	77.00 ±1.41 ^a^	92.00 ± 2.16 ^a^	
	T_4_	9.67 ± 0.47 ^b^	14.33 ± 0.47 ^c^	
	T_5_	12.00 ± 0.82 ^b^	16.67 ± 0.47 ^c^	
Chlorpyrifos	T_1_	0	0	Treatment: F = 5833.4, *p* < 0.001 Time: F = 312.5 *p* < 0.001 Treatment × time: F = 32.7, *p* < 0.001
	T_2_	77.33 ± 0,47 ^b^	86.67 ± 0.94 ^b^	
	T_3_	80.33 ± 0.47 ^a^	94.33 ± 1.70 ^a^	
	T_4_	10.33 ± 0.47 ^d^	17.67 ± 0.47 ^c^	
	T_5_	13.00 ± 082 ^c^	20.00 ± 0.82 ^c^	

T_1_ included ASB (100%), T_2_ featured a 0.5% insecticide concentration + 99.50% ATSB, T_3_ utilized a 1% insecticide concentration + 99% ATSB, T_4_ employed a 0.5% nano-formulated insecticide concentration + 99.50% ATSB, and T_5_ applied a 1% nano-formulated insecticide concentration + 99% ATSB. For each insecticide, means ± standard error followed by different lowercase letters (a,b,c,d) in a column indicate significant differences between treatments (*p* < 0.05).

**Table 3 insects-15-00070-t003:** Effect of commercial and nano-formulated carbamates on non-target adult *Coccinella septempunctata*.

Insecticides	Treatments	Mortality after 36 h (%)	Mortality after 72 h (%)	GLM Test Results
Carbosulfan	T_1_	0	0	Treatment: F = 4376.3, *p* < 0.001 Time: F = 112.3, *p* < 0.001 Treatment × time: F = 15.2, *p* < 0.001
	T_2_	81.00 ± 0.82 ^b^	86.00 ± 0.82 ^b^	
	T_3_	86.67 ± 1.25 ^a^	96.00 ± 1.41 ^a^	
	T_4_	13.67 ± 0.94 ^d^	20.00 ± 0.82 ^c^	
	T_5_	18.00 ± 0.82 ^c^	22.67 ± 0.94 ^c^	
Propoxur	T_1_	0	0	Treatment: F = 2833.5, *p* < 0.001 Time: F = 63.9, *p* < 0.001 Treatment × time: F = 6.1, *p* = 0.002
	T_2_	73.67 ± 2.7 ^b^	83.67 ± 2.36 ^b^	
	T_3_	81.33 ± 1.89 ^a^	88.33 ± 1.25 ^a^	
	T_4_	11.33 ± 0.47 ^c^	15.00 ± 0.82 ^c^	
	T_5_	13.67 ± 0.47 ^c^	18.33 ± 1.25 ^c^	
Methomyl	T_1_	0	0	Treatment: F = 3606.1, *p* < 0.001 Time: F = 114.3, *p* < 0.001 Treatment × time: F = 12.1, *p* < 0.001
	T_2_	71.00 ± 0.82 ^b^	84.00 ± 0.82 ^a^	
	T_3_	82.67 ± 2.05 ^a^	87.33 ± 2.49 ^a^	
	T_4_	10.00 ± 0.82 ^d^	17.33 ± 0.47 ^b^	
	T_5_	14.00 ± 0.82 ^c^	19.00 ± 0.82 ^b^	

T_1_ included ASB (100%), T_2_ featured a 0.5% insecticide concentration + 99.50% ATSB, T_3_ utilized a 1% insecticide concentration + 99% ATSB, T_4_ employed a 0.5% nano-formulated insecticide concentration + 99.50% ATSB, and T_5_ applied a 1% nano-formulated insecticide concentration + 99% ATSB. For each insecticide, means ± standard error followed by different lowercase letters (a,b,c,d) in a column indicate significant differences between treatments (*p* < 0.05).

## Data Availability

All data are contained within the article.

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
