# Peer review of "Assessment of Nano-Formulated Conventional Insecticide-Treated Sugar Baits on Mosquito Control and the Effect on Non-Target Aphidophagous Coccinella septempunctata"

_insects, 2024, doi:10.3390/insects15010070_

Round 1

Reviewer 1 Report

Comments and Suggestions for Authors

See comments and edits on the attached file.

Comments on the Quality of English Language

See file attached.

Author Response

Dear Professor

We would like to express our gratitude for the opportunity to improve the paper and appreciate your guidance. We have tried our best to ensure the required revising and corrections.

Reviewer 2 Report

Comments and Suggestions for Authors

Dear authors,

This is an interesting research study that assesses the efficacy of nano formulated conventional synthetic attractive sugar toxic baits against mosquitoes.

However, the study except for some minor revisions faces some major flaws in the methodology and data analysis. In more details:

Lines 76-86: This test seems redundant and could be omitted.

The mosquito species that were collected and used in the bioassays should be specified. Collection of mosquitoes and bioassays on mosquitoes in general is too poor information.

Also, it should be clarified for how many generations the collected adults and larvae were reared in the laboratory, i.e., the bioassays performed with Fxx generation adults.

Lines 211-225: The number of replicates (cages) in mosquito adults bioassays should be clearly stated.

From the graphs it seems that you used 3 replicates for each treatment combination, i.e., insecticide + concentration. The analysis of variance (ANOVA) for your data using only 3 replicates seems not appropriate. Please clarify if the basic requirements for ANOVA with your data are fulfilled, i.e., data homogeneity and normality, and provide the results of the relevant tests in the manuscript. Otherwise, you may consider non-parametric test for data analysis.

In figures 1,2 & 3 only the results from 36 and 72 hours are presented. Where are the results from “immediate” effect in the morning as stated in the materials and methods?

“Immediate” should be more precise, e.g., in 12 or 24 h post treatment.

For the results against the beetle, the of the overall ANOVA results should be presented in the text. I have the same concerns about the reliability of data analysis.

Tables 1,2&3: It should be written “For each insecticide, means ± standard error followed by different lowercase letters (a,b,c,d,e) in a column indicate significant differences between treatments (p < 0.05)”.

Lines 377-397: These paragraphs do not discuss the results so could be omitted.

Lines 433-435: There was not comparison of mortality between ATSB and N-ATSB so this statement is not supported from the data analysis and could be removed or rephrased.

Lines 495-497: Why “harmless”? Some mortality was observed, was omit this statement or rephrase it.

Author Response

Dear Professor

We would like to express our gratitude for the opportunity to improve the paper and appreciate your guidance. We have tried our best to ensure the required revising and corrections.Herein, we have listed the reviewer's comments and our response to them point-by-point.We are looking forward to hearing from you at your convenience.

Round 2

Reviewer 2 Report

Comments and Suggestions for Authors

Dear authors,

Thank you for considering my comments. However, the following points should be revisited and revise the manuscript accordingly.

Please clarify in the text how did you identify the collected mosquitoes Anopheles gambiae, e.g., the identification key you used. In the scope of the manuscript (Introduction) it should be clearly stated that the efficacy of nanoformulations was tested against An. gambiae. Also, in the legends of figures 1-3 An. gambiae should be mentioned, not generally mosquitoes.  

Tables 1, 2 & 3: Normally, the comparison of mortality was done for each insecticide at 36h and 72h separately (comparison for each column). If this is the case it should be stated “For each insecticide, means ± standard error followed by different lowercase letters (a,b,c,d,e) in a column indicate significant differences between treatments (p < 0.05)". In lines 308 & 323, it seems that mortality between 36 and 72h was compared, which is probably not correct. Please check and add in the text the result of the overall GLM test for at least some cases of treatment combinations, i.e., “insecticide x time of mortality check”.

Rephrase the first 4-5 lines of the discussion so that the discussion begins with your findings, not from other researchers.

Author Response

Dear professor

We would like to express our gratitude for the opportunity to improve the paper and appreciate your guidance. We have tried our best to ensure the required revising and corrections.

Herein, we have listed the reviewer's comments and our response to them point-by-point.

We are looking forward to hearing from you at your convenience.

Please clarify in the text how did you identify the collected mosquitoes Anopheles gambiae, e.g., the identification key you used. In the scope of the manuscript (Introduction) it should be clearly stated that the efficacy of nanoformulations was tested against An. gambiae. Also, in the legends of figures 1-3 An. gambiae should be mentioned, not generally mosquitoes.  

Response: Thank you for your comments; we have made all the suggested changes accordingly in the manuscript.

Tables 1, 2 & 3: Normally, the comparison of mortality was done for each insecticide at 36h and 72h separately (comparison for each column). If this is the case it should be stated “For each insecticide, means ± standard error followed by different lowercase letters (a,b,c,d,e) in a column indicate significant differences between treatments (p < 0.05)".

Response: Thank you for your comment; we have added the required changes for each insecticides in the Colum.

In lines 308 & 323, it seems that mortality between 36 and 72h was compared, which is probably not correct. Please check and add in the text the result of the overall GLM test for at least some cases of treatment combinations, i.e., “insecticide x time of mortality check”.

Response: Thank you for the important comment. We added the detailed statistical analysis according to the suggestions.

Rephrase the first 4-5 lines of the discussion so that the discussion begins with your findings, not from other researchers.

Response: Thank you for your comment; we have revised the paragraph in the discussion section.
